# Oral and rectal colonization of Gram-negative antimicrobial-resistant bacteria and Methicillin-resistant *Staphylococcus aureus* in one long-term care facility and changes in professional oral hygiene care: Cross-sectional and interventional study

Azusa Haruta[1]*, Mineka Yoshikawa[1], Maho Takeuchi[1], Miki Kawada-Matsuo[2,3], Mi Nguyen-Tra Le[4], Toshiki Kajihara[3,5], Yo Sugawara[5], Junzo Hisatsune[3,5], Hitoshi Komatsuzawa[2,3], Hiroki Ohge[3,6], Motoyuki Sugai[3,5], Kazuhiro Tsuga[1]

**1** Department of Advanced Prosthodontics, Hiroshima University Graduate School of Biomedical and Health Sciences, Hiroshima, Japan, **2** Department of Bacteriology, Hiroshima University Graduate School of Biomedical and Health Sciences, Hiroshima, Japan, **3** Project Research Center for Nosocomial Infectious Diseases, Hiroshima University, Hiroshima, Japan, **4** Faculty of Dentistry, University of Medicine and Pharmacy at Ho Chi Minh City, Ho Chi Minh City, Vietnam, **5** Antimicrobial Resistance Research Center, National Institute of Infectious Diseases, Tokyo, Japan, **6** Department of Infectious Diseases, Hiroshima University Hospital, Hiroshima, Japan

\* harutazusa@hiroshima-u.ac.jp

## Abstract

The spread of antimicrobial-resistant bacteria is a global threat. Our previous study investigated oral colonization by Gram-negative antimicrobial-resistant bacteria and methicillin-resistant *Staphylococcus aureus* (MRSA) in long-term care facilities. In this study, antimicrobial-resistant bacteria in the oral cavity of bedridden severely dependent elderly residents were investigated and the relationship of antimicrobial-resistant bacteria with oral and systemic status was clarified. In addition, the effect of professional oral care on antimicrobial-resistant bacteria colonization was investigated. This was a cross-sectional study followed by an interventional study. Fifty-seven residents were randomly assigned to screening for the presence of Gram-negative antimicrobial-resistant bacteria with 3rd-generation cephalosporin or carbapenem-resistance methods and the presence of oral and rectal MRSA using respective selective plates. Epidemiological data were collected from clinical records. The interventional study was conducted on 23 subjects who presented with Gram-negative antimicrobial-resistant bacteria or *S. aureus* in the cross-sectional study. The interventions included professional oral care once a week for 8 weeks in addition to daily oral care. Antimicrobial-resistant bacteria colonization before and after the intervention was compared. Among 57 subjects, antimicrobial-resistant bacteria were isolated from oral samples of 29 subjects and from rectal samples of 44 subjects. Among 29 subjects with oral

**Data availability statement:** All the data generated or analyzed in this study are included in this publication and its supplemental information file.

**Funding:** This study was supported by the Ministry of Health, Labour, and Welfare Japan (program Grant No. JPMH19HA1004 and JPMH22HA1002)(OH), MEXT Grants-in-Aid for Scientific Research (C) (Grant No. 20K10248)(MY), the Research Program on Emerging and Re-emerging Infectious Diseases from the Japan Agency for Medical Research and Development under Grant No. 21fk0108604j0001 and JP22fk0108604j0002(MS), and JSPS Grant-in-Aid for Research Activity Start-up (Grant No. 23K19745) (AH).

**Competing interests:** The authors have no conflicts of interest to declare.

**Abbreviations:** MRSA, methicillin-resistant *Staphylococcus aureus*, ARB, antimicrobial-resistant bacteria, NHCAP, Nursing and Health Care-Associated Pneumonia, LTCFs, long-term care facilities, ESBL, Extended-Spectrum β-Lactamase, GN-ARB, Gram-negative ARB, ADL, activity of daily living, PEGs, percutaneous endoscopic gastrotomy tubes, NGs, nasogastric tubes, FOIS, Food Oral Intake Scale, MALDI-TOF MS, matrix-assisted laser desorption/ionization-time-of-flight mass spectrometry, POT, PCR-based ORF typing, GCS, Glasgow Coma Scale, qSOFA, quick Sepsis Related Organ Failure Assessment.

antimicrobial-resistant bacteria, Gram-negative antimicrobial-resistant bacteria and MRSA were isolated from 21 and 17 subjects, respectively. Logistic regression analysis of the independent variables revealed that Non-oral nutritional intake was significantly related to Gram-negative antimicrobial-resistant bacteria positivity. In the interventional study, professional oral care failed to eliminate of oral antimicrobial-resistant bacteria in most subjects. This study showed that subjects with Non-oral nutritional intake had significantly higher rates of oral Gram-negative antimicrobial-resistant bacteria. These findings concluded that a short-term professional oral management has a limited capacity to eliminate antimicrobial-resistant bacteria. Our results provide important information for the control of infections caused by antimicrobial-resistant bacteria in the oral cavity.

## Introduction

In Japan, 29.0% of people were aged 65 years and over in 2022, and this percentage is projected to 38.7% in 2070. (2023 white paper by the Cabinet Office in Japan, Annual Report on the Ageing Society [Summary] FY2023). A long-term care insurance system that supports the anxiety and financial burden of long-term care for individuals is available in Japan. The percentage of people who are certified as requiring long-term care increases significantly after they reach 75 years of age. If support or long-term care is required due to systemic diseases, the applicant consults with the Older General Counseling Center and applies for certification for support or long-term care. After investigation and examination, the person is certified as requiring support levels one and two to nursing care levels one to five. There were 6,558,000 people certified as requiring long-term care in 2019, according to a report by the Ministry of Health, Labor and Welfare in Japan. Older people in need of care are more likely to develop pneumonia, which is the fifth leading cause of death in Japan. In particular, aspiration pneumonia accounts for a large proportion of deaths from pneumonia in elderly individuals. Oral bacteria are major causative pathogens for aspiration pneumonia and include antimicrobial-resistant bacteria (ARB) such as carbapenem-resistant *Enterobacterales*, *Acinetobacter baumannii* and *Pseudomonas aeruginosa* [1].

Japan has a unique medical care system, including long-term care insurance in addition to universal health insurance. The concept of Nursing and Health Care-Associated Pneumonia (NHCAP) arose because many older people experience aspiration pneumonia. According to the NHCAP guidelines of Japanese Respiratory Society, the population targeted by the NHCAP is at risk for aspiration pneumonia and underlying disease changes with age, requiring home care and admission to geriatric health facilities and long-term care facilities (LTCFs) [2]. It has been reported that ARB such as *P. aeruginosa*, MRSA, and Extended-Spectrum β-Lactamase (ESBL)-producing enterobacteria represent approximately 12% of the causative bacteria isolated from patients with NHCAP in Japan, and differences depending on the region and facility involved have been reported [2, 3].

It is speculated that the number of ARB will increase significantly in the future, and an action plan to combat ARB has been formulated in Japan. Infection control has been implemented extensively not only in medical facilities but also in LTCFs.

Oral colonization by Gram-negative ARB (GN-ARB) in 33% of subjects in LTCFs was found to be associated with relatively high activity of daily living (ADL) in 2020 [4]. The patient's ability to performed ADL varied, and some of the subjects were outpatients undergoing rehabilitation. Additionally, our previous study conducted a similar study at multiple LTCFs and reported that oral colonization by GN-ARB occurred in 17.4% (16.2% and 19.4%, respectively) of the subjects in 6 LTCFs, including the Welfare Facility for the Elderly Requiring Long-term Care and Geriatric Health Services Facility, in 2023 [5]. Therefore, this study aimed to confirm the presence or absence of ARB in the oral cavity in bedridden severely dependent older people and to clarify the relationship between the oral and swallowing functions of the subjects and the results of the colonization survey after 8 weeks of professional oral care.

## Materials and methods

### Cross-sectional study

**Study design and subjects.** This study was performed to assess the prevalence of ARB in the oral cavity among bedridden, severely dependent older adults in an integrated facility for medical and long-term care located in Hiroshima city (Facility X). This study was approved by the ethical committee of the Hiroshima University Hospital Review Board (E-1740). All procedures were performed in accordance with institutional guidelines. Subjects were prospectively selected from 138 residents at Facility X from November 25th, 2020 to June 30th, 2021. After random sampling, the cooperation was requested with the permission of the family. Informed consent was obtained from the subject's families.

The sample size was calculated for the primary analysis on the basis of differences observed in a previous study [4]. In this study, 16.2% of the ARB-positive subjects had percutaneous endoscopic gastrotomy tubes (PEGs). Assuming an $\alpha$ level of 0.05 and a statistical power of 80%, the required number of subjects for each group was 23. To allow for study drop-outs, 57 residents were randomly assigned.

**Inclusion criteria.** This study included subjects who were over 65 years old and who were certified as having a Care Level of four or five in the Japanese LTC insurance system. At level four, full assistance is required in all aspects of daily life, such as eating, excretion, and bathing, and simultaneously, communication becomes difficult and problematic behavior that interferes with daily life is frequently observed. Level five in most cases applies to older adults who are bedridden and completely unable to communicate. Also, subjects who had dysphagia, i. e., subjects who were receiving a diet for dysphagia or alternative nutrition were included because they had a high risk of aspiration pneumonia and NHCAP [6, 7].

In addition, subjects who lived in Facility X for more than six months and who had no access to other medical facilities were included.

**Exclusion criteria.** The subjects who received antibiotic prescription more than twice within 90 days, according to the Japanese NHCAP risk factors were excluded [2, 3].

**Epidemiological investigation.** The following data were collected from the medical records: age, sex, care level, systemic disease such as cerebrovascular disorder, cardiac disease and dementia (for example, Alzheimer's disease, vascular dementia, Lewy body dementias and frontotemporal dementia). If these diseases were diagnosed, the subjects were considered to have cognitive decline. In addition, the presence or absence of medical devices, such as urinary catheters, PEGs, and nasogastric tubes (NGs) was checked. ADL levels according to the Barthel Index [8, 9], nutritional intake according to the Food Oral Intake Scale (FOIS) [10], and the residential floor of subjects were identified as risk factors for oral colonization by ARB. The FOIS is a seven-point ordinal scale that estimates and documents changes in functional eating abilities. The subjects with a nutritional method of the Non-oral food intake were regarded as FOIS level one (i. e., nothing by mouth). Subjects who were tube dependent

with minimal attempts to consume food or liquid orally were regarded as level two, and subjects who were tube dependent with consistent oral intake of food or liquid were deemed as level three. Subjects with a FOIS level from four to seven had started a total oral diet. The nutritional methods used for FOIS levels one to three, comprising subjects who were tube dependent, were separated into PEG or NG. The medical records were accessed for research purposes from December 9th, 2020 to June 30th, 2021 and had access to information on medical records that could identify individual subjects during or after data collection.

Our cross-sectional study defined subjects with ARB as having either GN-ARB, which was selected for by screening selective plates, or MRSA, which was selected for by PCR screening for the presence of the gene.

**Sample collection and microbiological methods.** To check the ARB carriage, oral swabs were collected from all subjects and spread directly onto CHROMagar ™ ESBL, CHROMagar ™ mSuperCARBA ™, CHROMagar ™ Candida plates (Kanto Chemical, Japan) and staphylococcal selective media (Nissui Pharmaceutical, Tokyo, Japan). The rectal samples were collected via the same methods.

CHROMagar ™ ESBL, CHROMagar ™ mSuperCARBA ™ and CHROMagar ™ Candida plates were incubated for 18−24 hours at 37 °C under aerobic conditions, and colonization was observed after incubation. To determine resistance, all positive colonies were incubated on new plates. Those that grew on the CHROMagar ™ ESBL and CHROMagar ™ mSuperCARBA ™ plates but not on CHROMagar ™ Candida plates were considered GN-ARB. Using PCR, the presence of ESBL-producing genes ($bla_{CTX-M-group-1}$, $bla_{CTX-M-group-2}$, $bla_{CTX-M-group-8}$, $bla_{CTX-M-group-9}$, $bla_{TEM}$, and $bla_{SHV}$) and carbapenemase-producing genes ($bla_{GES}$, $bla_{IMP}$, and $bla_{NDM}$) were confirmed, and the primers used in this study are listed in S1 Table. Additionally, strain identification was confirmed via matrix-assisted laser desorption/ionization-time-of-flight mass spectrometry (MALDI-TOF MS; MALDI Biotyper, Bruker Daltonics, Billerica, USA). The colonies on CHROMagar ™ ESBL were collected as third-generation cephalosporin-resistant bacteria and on CHROMagar ™ mSuperCARBA ™ as carbapenem-resistant bacteria.

Staphylococcal selective medium plates were also incubated for 2 days at 37 °C under aerobic conditions, after which individual yellow colonies (up to 4 colonies in each sample) were picked and replated. DNA extraction for PCR and PCR was conducted as described in a previous study [11]. Using the above method, ARB was screened, third-generation cephalosporin-resistant bacteria, carbapenem-resistant bacteria, and MRSA.

## Interventional study

**Overview of the interventional study.** This was a cross-sectional study followed by an interventional study (Fig 1). Of those subjects who had GN-ARB or *S. aureus* in the earlier cross-sectional study, 23 provided informed consent (Fig 2).

The flowchart of the interventional study is shown in Fig 3. The interventions involved professional oral care once a week for 8 weeks by dentists and dental hygienists who work full-time at the facility in addition to daily oral care. Professional oral management in our study included the use of toothbrushes, interdental brushes, floss, and oral moisturizers. The present study did not use mouthwash because the disinfectants in mouthwash could have affected the presence or absence of resistant bacteria [12]. The intervention technique employed, as well as the oral cleaning products and clothing, was standardized among examiners.

The intervention group was divided into 4 groups (Groups 1–4) and conducted the interventions during each period. After the cross-sectional study, the residents in Group 1 were subjected to the intervention (professional oral management). In Groups 2–4, the intervention was started more than 3 months after the cross-sectional study, and the presence of oral ARB was investigated again. Also, the frequency of ARB was investigated within 2 weeks after the intervention. The presence or absence of GN-ARB and *S. aureus* including MRSA were determined, using the same methods as in the cross-sectional study. The bacterial species of GN-ARB strains were examined before and immediately after the intervention (1–2 weeks after the intervention ended). Additionally, *S. aureus* typing was examined by PCR-based ORF typing (POT) using a Cica Geneus® Staph POT Kit (Kanto Chemical, Japan) [13, 14].

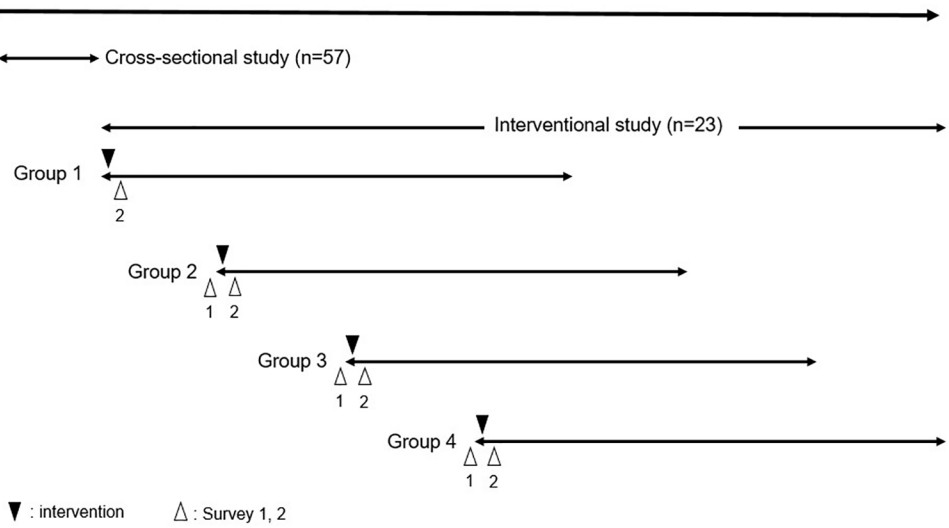

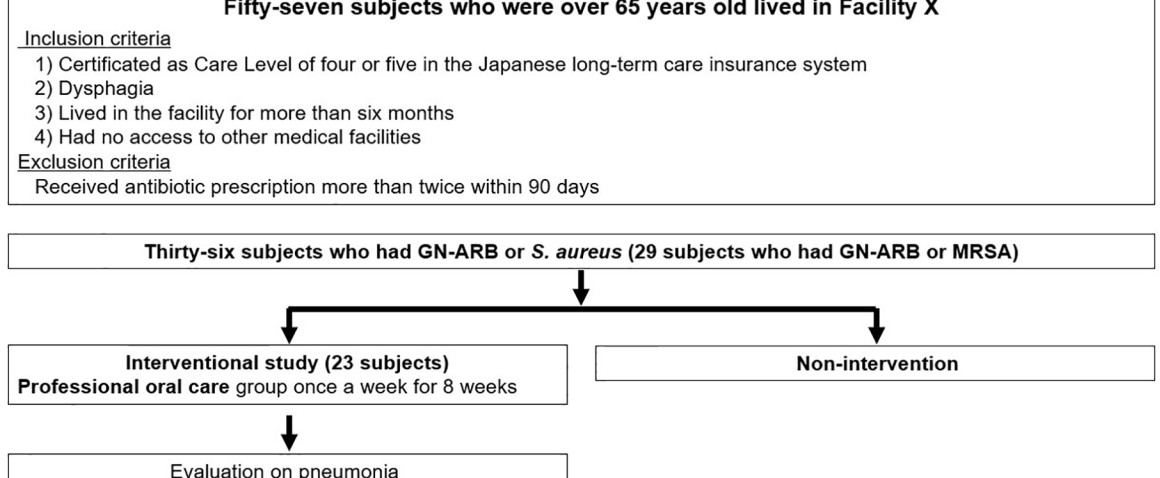

**Fig 1. Overall flow of the study.** This study performed a cross-sectional study followed by an interventional study. The intervention population were divided into 4 groups (Groups 1-4) and conducted the interventions during each period. After the cross-sectional study, Group 1 was subjected to intervention. In Groups 2 to 4, the presence of oral ARB was investigated again before the intervention.

**Fig 2. Flowchart of the study.** This study selected fifty-seven subjects over 65 years of age according to the criteria for this cross-sectional study. For those who had GN-ARB or *S. aureus* in the earlier cross-sectional study, an intervention study was conducted on those subjects who provided informed consent.

**Evaluation of pneumonia.** The occurrence of pneumonia in the subjects in the interventional group was examined from immediately to one year after the intervention (Fig 3). Since it was difficult to perform uniform chest radiographs and blood draws on patients with suspected pneumonia at the facility, the occurrence of pneumonia was determined based on vital signs and physical findings. The Glasgow Coma Scale (GCS) score was checked, disturbance in consciousness, respiratory rate, blood pressure, pulse and body temperature as vital signs. The onset of pneumonia was defined as

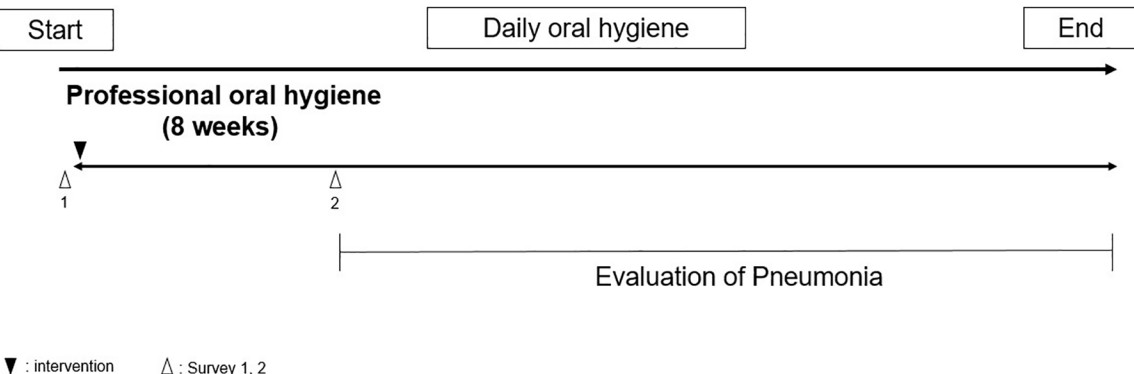

▼ : intervention        △ : Survey 1, 2

* Survey 1, before the intervention; Survey 2, immediately after the intervention within 1 to 2 weeks

**Fig 3. Flowchart of the intervention study.** The intervention was performed for eight weeks and the survey was applied before and after the intervention. The pneumonia was evaluated from the end to one year after the intervention.

when two out of the following four items were met: respiratory rate>29 times per minute, presence of sputum, a body temperature of 38 degrees or higher and presence of a rale. The severity of pneumonia was evaluated using the CRB-65 and quick Sepsis Related Organ Failure Assessment (qSOFA) [15, 16].

If there was onset of pneumonia, the medical records were searched for the number of days since onset and the number of occurrences from April 29th, 2021 to March 5th, 2023.

Additionally, the disease severity was assessed using the CRB-65 score or qSOFA and determined the treatment classifications based on the Japanese NHCAP guidelines [16–19].

This study was approved by the ethical committee of the Hiroshima University Hospital Review Board on December 28th, 2020 (C-304-1). All procedures were conducted in accordance with institutional guidelines. Written informed consent was obtained from the subject's families due to an impaired mental capacity to consent. All study protocols were conducted in accordance with the principles of the Declaration of Helsinki. The information on medical records that could identify individual participants had been accessed during or after data collection.

**Statistical analysis.** Univariate statistical methods were used to compare the baseline clinical data between the subjects who had oral colonization by ARB and those who did not. Associations were analyzed using Fisher's exact test. Then, logistic regression analysis was conducted using all variables with a P value < 0.05, which was considered to indicate statistical significance. Additionally, comparisons of clinical outcomes stratified by the presence of oral ARB (in a cross-sectional study) before and after the intervention were performed using Fisher's exact test. All statistical analyses were performed using JMP® Pro 15.0.0 statistical software (SAS Institute Inc., Cary, NC, USA).

## Results

### Cross-sectional study

**Subject epidemiology.** Fifty-seven of the 138 older adults participated in this study. The mean age was $87.14 \pm 7.67$ years (69–102 years). Seven subjects were men (12.3%), with a mean age of $84.86 \pm 7.13$ years (75–95 years) and 50 subjects were women (87.7%), with a mean age of $87.46 \pm 7.75$ years (69–102 years).

**Isolation of ARB from oral and rectal samples.** Among 57 subjects, ARB were isolated from oral samples collected from 29 subjects (50.9%) and from rectal samples collected from 44 subjects (77.2%) (Table 1). Among 29 subjects with oral ARB, GN-ARB and MRSA were isolated from 21 (36.8%) and 17 subjects (29.8%), respectively. Both GN-ARB and MRSA were isolated from the oral cavity of 10 subjects (17.5%). Among the 44 subjects with rectal

**Table 1. Number of subjects with resistant bacteria in the cross-sectional study.**

| | Subjects, n = 57 | |
|---|---|---|
| | n | (%) |
| Subjects with Oral resistant bacteria | | |
| Resistant bacteria (Gram-negative, MRSA) | 29 | (50.9) |
| Both Gram-negative bacteria and MRSA | 10 | (17.5) |
| Gram-negative resistant bacteria | | |
| Any isolates | 21 | (36.8) |
| ESBL-producing Enterobacteriaceae | 11 | (19.3) |
| *P. aeruginosa* | 5 | (8.8) |
| MRSA | 17 | (29.8) |
| Subjects with Rectal resistant bacteria | | |
| Gram-negative bacteria | | |
| Any isolates | 44 | (77.2) |
| ESBL-producing Enterobacteriaceae | 34 | (59.6) |
| *P. aeruginosa* | 15 | (26.3) |
| Gram-negative resistant bacteria from both the oral cavity and rectum | 18 | (31.6) |
| Same species in the oral cavity and rectum (Gram-negative) | 11 | (19.3) |

MRSA, methicillin-resistant *Staphylococcus aureus* (*S. aureus*);

ESBL, extended-spectrum β-lactamase.

ARB, only GN-ARB was isolated. Thirty-one GN-ARB were isolated from the oral cavity of 21 subjects and 60 from the rectum of 44 subjects (Table 2). The major oral bacterial species were *Acinetobacter ursingii* (8 isolates), followed by *Pseudomonas aeruginosa* (5 isolates), *E. coli* (5 isolates) and *Stenotrophomonas maltophilia* (5 isolates), while the major rectal bacterial species were *Escherichia coli* (32 isolates), followed by *P. aeruginosa* (15 isolates) and *Proteus mirabilis* (5 isolates) (Table 2).

GN-ARB was isolated from both the oral cavity and rectum of 18 subjects (31.6%), while MRSA was only isolated from the oral cavity. Two or three bacterial species of GN-ARB in one subject were isolated from the oral cavity of 6 subjects and from the rectum of 14 subjects (S2 Table). Among the 44 subjects with GN-ARB, 11 had the same bacterial species of GN-ARB isolated from the oral cavity and rectum (Table 1).

**Identification of ESBL genes in GN-ARB strains of ARB.** To check the ESBL genes, PCR was performed using each specific primer to detect the respective ESBL gene. This study identified 11 (35.5%) of the 31 oral GN-ARB isolates and 18 (30.5%) of the 59 rectal GN-ARB isolates (Table 2). Among them, 9 and 37 ESBL-producing bacteria were isolated from the oral cavity and rectum, respectively by PCR (S3 Table). The most common gene identified in both the oral and rectal isolates was *bla*$_{CTX-M-group-9}$ (5/9, 25/37), and all the isolates carrying this gene were *E. coli* (Table 3). Ten isolates had more than one ESBL gene per isolate (2 oral isolates and 8 rectal isolates).

**Demographics and clinical characteristics of subjects who were carries of ARB.** The demographic and clinical characteristics of all the subjects were divided into two groups, ARB-positive and ARB-negative (Table 4). Next, this study analyzed the relationships of MRSA and GN-ARB with the clinical characteristics. The subjects with Non-oral food intake (OR 37.2, 95% CI 7.938–174.341, P < 0.0001) showed greater GN-ARB colonization in the oral cavity (Table 4), but no significant differences in the presence of MRSA (Table 5). Age, sex, ADL according to the Barthel Index, the facility's ward, and comorbidities were not risk factors for ARB oral colonization. Logistic regression analysis using these factors as independent variables revealed that Non-oral food intake was significantly related to GN-ARB positivity in severe-level care-needs residents in LTCFs (Table 6).

## Table 2. Number of subjects with oral and rectal isolates.

| | Subjects with Oral isolates n=57 | | Subjects with Rectal isolates n=57 | |
|---|---|---|---|---|
| | n | (%) | n | (%) |
| Gram-negative resistant bacteria | | | | |
| Acinetobacter baumanii | 0 | (0.0) | 1 | (1.8) |
| Acinetobacter ursingii | 8 | (14.0) | 0 | (0.0) |
| Citrobacter freundii | 0 | (0.0) | 1 | (1.8) |
| Escherichia coli | 5 | (8.8) | 32 | (56.1) |
| Klebsiella | | | | |
| K. oxytoca | 1 | (1.8) | 1 | (1.8) |
| K. pneumoniae | 1 | (1.8) | 3 | (5.3) |
| Proteus mirabilis | 3 | (5.3) | 5 | (8.8) |
| Pseudomonas | | | | |
| P. aeruginosa | 5 | (8.8) | 15 | (26.3) |
| Other Pseudomonas | 2 | (3.5) | 1 | (1.8) |
| Serratia marcescens | 1 | (1.8) | 1 | (1.8) |
| Stenotrophomonas maltophilia | 5 | (8.8) | 0 | (0.0) |
| ESBL-producing gene | 11 | (19.3) | 18 | (31.6) |
| $bla_{CTX-M-group-1}$ | 1 | (1.8) | 4 | (7.0) |
| $bla_{CTX-M-group-2}$ | 3 | (5.3) | 3 | (5.3) |
| $bla_{CTX-M-group-9}$ | 5 | (8.8) | 23 | (40.4) |
| $bla_{TEM}$ | 2 | (3.5) | 9 | (15.8) |
| $bla_{SHV}$ | 1 | (1.8) | 2 | (3.5) |
| Staphylococcus aureus | 27 | (47.4) | 18 | (31.6) |
| MRSA | 17 | (29.8) | - | |

## Table 3. Number of isolates with ESBL-producing gene.

| Bacterial species | Isolation site* | n | $bla_{CTX-M-group-1}$ | $bla_{CTX-M-group-2}$ | $bla_{CTX-M-group-8}$ | $bla_{CTX-M-group-9}$ | $bla_{TEM}$ | $bla_{SHV}$ |
|---|---|---|---|---|---|---|---|---|
| A. baumannii | R | (1) | 0 | 0 | 0 | 0 | 0 | 0 |
| A. ursingii | O | (8) | 0 | 0 | 0 | 0 | 0 | 0 |
| E. coli | O | (5) | 0 | 0 | 0 | 5 | 1 | 0 |
| | R | (32) | 4 | 0 | 0 | 25 | 8 | 0 |
| K. oxytoca | O | (2) | 0 | 0 | 0 | 0 | 0 | 0 |
| | R | (1) | 0 | 0 | 0 | 0 | 0 | 0 |
| K. pneumoniae | O | (1) | 1 | 0 | 0 | 0 | 1 | 1 |
| | R | (2) | 2 | 0 | 0 | 0 | 2 | 2 |
| P. aeruginosa | O | (12) | 0 | 0 | 0 | 0 | 0 | 0 |
| | R | (17) | 0 | 0 | 0 | 0 | 0 | 0 |
| P. mirabilis | O | (3) | 0 | 3 | 0 | 0 | 0 | 0 |
| | R | (2) | 0 | 2 | 0 | 0 | 0 | 0 |
| S. maltophilia | O | (8) | 0 | 0 | 0 | 0 | 0 | 0 |

* O, oral cavity; R, rectum.

**Table 4. Clinical characteristics of subjects and risk factors associated with oral colonization by ARB Gram-negative.**

| Variable | Residents with GN-ARB-positive, n=21 | Residents with GN-ARB-negative, n=36 | Univariate analysis | | P value |
|---|---|---|---|---|---|
| | n (%) | n (%) | OR | 95% CI | |
| Age over 90 years, n(%) | 7 (33.3) | 11 (30.6) | 1.136 | 0.359–3.594 | 1.0 |
| BI score 0, n(%) | 19 (90.5) | 25 (69.4) | 4.18 | 0.827–21.131 | 0.103 |
| Ward, n(%) | | | | | |
| 4th floor | 8 (38.1) | 13 (36.1) | 1.089 | 0.358–3.313 | 1.0 |
| 3rd floor | 8 (38.1) | 8 (22.2) | 2.154 | 0.662–7.011 | 0.232 |
| 1st floor | 5 (23.8) | 15 (41.7) | 0.438 | 0.131–1.457 | 0.251 |
| Non-oral food intake | 18 (85.7) | 5 (13.9) | 37.2 | 7.938–174.341 | < 0.0001 [a, **] |
| PEG feeding | 10 (47.6) | 2 (5.56) | 15.455 | 2.928–81.562 | 0.0003 [a, **] |
| NG tube feeding | 8 (38.1) | 3 (8.33) | 6.769 | 1.55–29.558 | 0.012 [a,*] |

GN-ARB, Gram-negative antimicrobial-resistant bacteria; OR, Odds ratio; CI, Confidence interval; BI, Barthel Index; PEG, Percutaneous endoscopic gastrotomy tube; NG, Nasogastric.

[a]Fisher's Exact Test.

*P value <0.05, **P value <0.01.

**Table 5. Clinical characteristics of subjects and risk factors associated with oral colonization by ARB MRSA.**

| Variable | Residents with MRSA -positive, n=16 | Residents with MRSA -negative, n=41 | Univariate analysis | | P value |
|---|---|---|---|---|---|
| | n (%) | n (%) | OR | 95% CI | |
| Age over 90 years, n(%) | 6 (37.5) | 12 (29.3) | 1.45 | 0.43–4.888 | 0.545 |
| Male, n(%) | 3 (18.8) | 4 (9.76) | 2.135 | 0.420–10.839 | 0.388 |
| BI score 0, n(%) | 13 (81.3) | 31 (75.6) | 1.398 | 0.33–5.921 | 0.740 |
| Ward, n(%) | | | | | |
| 4th floor | 10 (62.5) | 11 (26.8) | 4.545 | 1.335–15.48 | 0.017 [a, *] |
| 3rd floor | 3 (18.8) | 13 (31.7) | 0.497 | 0.120–2.051 | 0.513 |
| 1st floor | 3 (18.8) | 17 (41.5) | 0.326 | 0.08–1.322 | 0.132 |
| Non-oral food intake | 9 (56.3) | 14 (34.2) | 2.48 | 0.762–8.069 | 0.145 |
| PEG feeding | 3 (18.8) | 9 (22.0) | 0.821 | 0.191–3.523 | 1.000 |
| NG tube feeding | 6 (37.5) | 5 (12.2) | 4.32 | 1.089–17.14 | 0.057 |

ARB, Antimicrobial-resistant bacteria; OR, Odds ratio; CI, Confidence interval;

BI, Barthel Index; PEG, Percutaneous endoscopic gastrotomy tube; NG, Nasogastric.

[a]Fisher's Exact Test

*P value <0.05, **P value <0.01

**Table 6. Results of logistic regression analysis predicting oral colonization by ARB GN-ARB.**

| Item | B | Standard Error | Chi square | Degree of freedom | P value | 95% CI |
|---|---|---|---|---|---|---|
| Non-oral food intake | 1.972 | 0.454 | 16.12 | 2 | <0.0001 | 1.103–3.083 |
| PEG feeding | 0 | 0 | – | 2 | – | – |
| NG tube feeding | −0.314 | 0.514 | 0.37 | 2 | 0.541 | −2.417–0.691 |
| Constant | −0.677 | 0.454 | 2.23 | 3 | 0.136 | −1.605–0.241 |

PEG, Percutaneous endoscopic gastrotomy tube; NG, Nasogastric.

## Interventional study

The present study investigated the effect of professional oral care on the elimination of ARB in the oral cavity.

**GN-ARB-carriage survey.** In the intervention group, four subjects (#2, 8, 30, 56) who received oral GN-ARB before the intervention had no GN-ARB after the intervention (Table 7). Conversely, five subjects (#16, 17, 29, 41, 53) with no GN-ARB before the intervention had GN-ARB after the intervention. In 6 subjects (#1, 3, 4, 5, 33, 37), the same GN-ARB bacterial species were detected before and after the intervention. In two subjects (#38 and 39 different bacterial species were found before and after the intervention.

***Staphylococcus aureus*-carriage survey.** Three subjects (#39, 41, 56) with *S. aureus* (two with MRSA) before the intervention had no *S. aureus* after the intervention (Table 7). Conversely, sixteen subjects (#1, 3, 4, 5, 6, 10, 13, 16, 17, 29, 30, 33, 37, 40, 53, and 63) had *S. aureus* both before and after the intervention, and twelve of them (#4, 6, 10, 13, 16,

**Table 7. The colonization of ARB in the oral cavity before and after the intervention.**

| Subject | Group | | intervention | | Subject | Group | | intervention | |
| | | | before | after 4W | | | | before | after 4W |
|---|---|---|---|---|---|---|---|---|---|
| 1 | 1 | GN-ARB | Pm | Pm | 30 | 2 | GN-ARB | Sm | – |
| | | Sa | MR(93-157-30) | MR(75-211-32) | | | Sa | MS(4-50-96) | MS(4-50-96) |
| 2 | 1 | GN-ARB | Sm | – | 31 | 2 | GN-ARB | – | – |
| | | Sa | – | – | | | Sa | – | – |
| 3 | 1 | GN-ARB | Pm | Pm | 33 | 3 | GN-ARB | Pa, Ko | Pa, Ko, Sm |
| | | Sa | MS(14-63-64) | MR(106-167-96) | | | Sa | MR(106-167-96) | MR(106-167-96) |
| 4 | 1 | GN-ARB | Pm | Pm | 37 | 3 | GN-ARB | Pa | Pa, Sm |
| | | Sa | MR(106-191-125) | MR(106-191-125) | | | Sa | MR(106-167-96) | MR(106-167-96) |
| 5 | 1 | GN-ARB | Ec | Ec, Ab | 38 | 3 | GN-ARB | Sm | Au |
| | | Sa | MS(0-31-1) | MS (8 –3 -1) | | | Sa | – | MS(4-127-104) |
| 6 | 1 | GN-ARB | – | – | 39 | 3 | GN-ARB | Ec, Sm | Pa, Pm |
| | | Sa | MR(93-137-2) | MR(93-137-2) | | | Sa | MR(106-167-96) | – |
| 8 | 1 | GN-ARB | Au | – | 40 | 3 | GN-ARB | – | – |
| | | Sa | – | – | | | Sa | MS(2-57-80) | MS(2-57-80) |
| 10 | 1 | GN-ARB | – | – | 41 | 3 | GN-ARB | – | Sm |
| | | Sa | MS(10-167-96) | MS(10-167-96) | | | Sa | MR(106-167-96) | – |
| 13 | 2 | GN-ARB | – | – | 53 | 4 | GN-ARB | – | Pm |
| | | Sa | MS(40-26-0) | MS(40-26-0) | | | Sa | MR(106-167-96) | MR(106-137-80) |
| 16 | 1 | GN-ARB | – | Ab | 56 | 4 | GN-ARB | Pa, Kp | – |
| | | Sa | MS(2-27-53) | MS(2-27-53) | | | Sa | MS(4-59-80) | – |
| 17 | 2 | GN-ARB | – | Au | 63 | 4 | GN-ARB | – | – |
| | | Sa | MR(98-179-101) | MR(98-179-101) | | | Sa | MR(106-167-96) | MR(106-167-96) |
| 29 | 2 | GN-ARB | – | Au | | | | | |
| | | Sa | MS(4-111-125) | MS(4-111-125) | | | | | |

Cases in which resistant bacteria were detected before intervention but not detected after intervention are shaded blue. Cases in which the same bacterial species (GN-ARB) or type (*S. aureus*) were detected before and after the intervention are shaded gray.

GN-ARB, Gram-negative antimicrobial-resistant bacteria;

Sa, *Staphylococcus aureus;*

Pm, *P. mirabilis*; Sm, *S. maltophilia*; Ec, *E. coli*; Ab, *A. baumanii*;

Au, *A. ursingii*; Pa, *P. aeruginosa*; Ko, *K. oxytoca*; Kp, *K. pneumoniae*;

MS, Methicillin-susceptible *S. aureus*; MR, Methicillin-resistant *S. aureus*

17, 29, 30, 33, 37, 40, and 63) had the same type of *S. aureus* both before and after the intervention. Eight subjects (#3, 10, 33, 37, 39, 41, 53, and 63) had the same type of MRSA (106-167-96) either before or after the intervention, or both before and after the intervention.

**Pneumonia and other clinical outcomes.** In the interventional group, seven subjects had pneumonia during the observation period. This study compared the rates of GN-ARB and *S. aureus* carriage in the oral cavity before and after the intervention between subjects who had pneumonia and those who did not have pneumonia (Table 8), and no significant difference were detected.

One subject with eight times episodes of pneumonia and a history of urinary tract infections, had a cause of death due to refractory pneumonia. This subject had a history of cerebral infarction and received nutrition via gastrostomy. EBSL-producing *E. coli* were isolated from the oral cavity and rectum of this subject in the cross-sectional study and four bacterial species of GN-ARB from the oral cavity of this subject both before and after the intervention.

Additionally, this study recognized that some subjects developed infections such as urinary tract infections and sepsis, in addition to pneumonia. Therefore, the carriage rates of GN-ARB and *S. aureus* were compared before and after intervention between subjects with and without infectious diseases (Table 8), and no significant differences were found. The mortality rates were also compared between the two groups, and no significant differences were identified.

## Discussion

This study found that the percentage of GN-ARB detected in the oral cavity or rectum was 36.8% or 77.2%, respectively. Our previous study reported the detection of oral or rectal GN-ARB respectively in 17.4% or 54.5% of residents of welfare facilities for older Requiring Long-term Care and Geriatric Health Services Facilities [5]. Compared to previous reports [5], the isolation rate from both the oral cavity and rectum in this study was greater in the present analysis, while it was almost the same

**Table 8. Comparison of oral ARB of carriage rates both before and after intervention.**

| | Both | | | Same species[b] | | |
|---|---|---|---|---|---|---|
| | Positive, n = 8 | Negative, n = 15 | | Positive, n = 6 | Negative, n = 17 | |
| GN-ARB | n | (%) | n | (%) | p-value[a] | n | (%) | n | (%) | p-value[a] |
| Onset of pneumonia | 4 | (50.0) | 3 | (20.0) | 0.18 | 2 | (33.3) | 5 | (29.4) | 1 |
| History of infection diseases | 5 | (62.5) | 5 | (33.3) | 0.22 | 3 | (50.0) | 7 | (41.2) | 1 |
| Death | 4 | (50.0) | 5 | (33.3) | 0.66 | 2 | (33.3) | 7 | (41.2) | 1 |
| | Both | | | Same type[b] | | |
| | Positive, n = 14 | Negative, n = 9 | | Positive, n = 9 | Negative, n = 14 | |
| *S. aureus* | n | (%) | n | (%) | p-value[a] | n | (%) | n | (%) | p-value[a] |
| Onset of pneumonia | 4 | (28.6) | 3 | (33.3) | 1 | 2 | (22.2) | 5 | (35.7) | 0.66 |
| History of infection diseases | 5 | (35.7) | 5 | (55.6) | 0.42 | 3 | (33.3) | 7 | (50.0) | 0.67 |
| Death | 3 | (21.4) | 6 | (66.7) | 0.077 | 2 | (22.2) | 7 | (50.0) | 0.23 |
| | Both | | | Same type[b] | | |
| | Positive, n = 6 | Negative, n = 17 | | Positive, n = 5 | Negative, n = 18 | |
| MRSA | n | (%) | n | (%) | p-value[a] | n | (%) | n | (%) | p-value[a] |
| Onset of pneumonia | 2 | (33.3) | 5 | (29.4) | 1 | 1 | (20.0) | 6 | (33.3) | 1 |
| History of infection diseases | 2 | (33.3) | 8 | (47.1) | 0.66 | 1 | (9.1) | 9 | (75.0) | 0.34 |
| Death | 2 | (33.3) | 7 | (41.2) | 1 | 1 | (9.1) | 8 | (66.7) | 0.61 |

[a]Fischer's exact test

[b]Same bacterial species or type between before and after the intervention

as the rate reported by Le et al. [4]. Compared to subjects in previous reports, all of the subjects in the present study were in the terminal stages of dementia who were bedridden at all times. This information suggests that the prevalence of resistant bacteria increases with the terminal stage of disease. Kajihara et al. reported a higher percentage of *P. aeruginosa* carriage that affecting survival (8.8% in the oral cavity and 26.3% in the stool in this study) compared to the isolation rates reported by Kajihara et al. and Le et al. (3.4% in the oral cavity, 2.8% in the stool, and 4% in the oral cavity). Higher rates of *P. aeruginosa* carriage was observed in terminally ill patients, indicating a poorer chance of survival. Many tissues are involved in the complicated movements involved in oral function, and they interact together with mediation of the saliva when we eat or speak. Therefore, even if ARB invades the oral cavity, the ARB may be swallowed and pass into the stomach with the saliva or swallowed at mealtime by oropharyngeal movements as a self-cleaning mechanism. The subjects in the present study, however, had significantly reduced oropharyngeal function, and it is possible that ARB would not disappear with daily oral care alone. None of the subjects developed NHCAP during our research, but the risk of developing NHCAP was considerable if the oral cavity was extremely dirty, oral care was not available, the amount of oral ARB increased, and physical fitness decreased. A Cochrane review by Liu et al. [20] reported that specialized oral hygiene management may reduce the mortality rate of patients compared to routine care, but no strong evidence was found. In addition, Myolotte et al. [21] reported that it is uncertain whether oral hygiene is a risk factor for NHCAP and whether improving oral hygiene reduces the incidence of infections, because it is difficult to distinguish between viral pneumonia and aspiration pneumonia. Therefore, it will be necessary to focus on bacterial infections and conduct detailed examinations in the future.

In the present study, all the subjects were in a chronic bedridden condition, were in the terminal stage of dementia, and had many comorbidities. Our previous report in 2020 report on ARB in the LTCF showed that ARB [4] were released from the oral cavity. Previously, Vink et al. reported some reports examined the presence of ARB in the oropharynx, but most other reports confirmed the presence of ARB colonization from the stool or rectum [22], and few studies examining the mouth have been performed. The possibility of transmission from diapers and stools between staff and patients and that staff themselves can be carriers has also been reported [23, 24].

Doctors and nurses were stationed at the facility that performed our study, and appropriate infection control measures were followed. The subjects received the same quality of medical care such as excretion care and oral care. In addition, dental hygienists in the facility determined the condition of the oral cavity and function, such as the number of remaining teeth and use of dentures among others. This professional care was provided throughout the year, although the intervals varied. Daily oral care was delivered twice a day by care givers or nurses. For subjects with severe oral hygiene conditions, three rounds of daily oral care were provided. Additionally, the registered dietitians in the facility performed nutritional management. The eating and swallowing abilities of the subjects were assessed using screening tests and/or videofluorography. However, even in such a well-organized environment, ARB colonization of the oral cavity was detected. In Japan, integrated facilities for medical and long-term care have medical functions such as nursing care and terminal care, as well as functions to serve as a living facility, in addition to daily medical management for people requiring long-term care. A countermeasure guide was issued in Japan in 2023 (https://www.mhlw.go.jp/content/10900000/001168459.pdf), and appropriate infection countermeasures and appropriate antibiotic prescriptions are being promoted.

This study found that subjects with Non-oral food intake had significantly greater rates of oral GN-ARB. Previous studies have reported that the oral microbiome of patients with Non-oral food intake differ from that of patients with oral food intake [25, 26]. As Non-oral intake is a factor for oral dryness, the oral environment of patients with Non-oral intake is different from that of patients with oral intake [27]. Therefore, this study suggests that the oral cavity of subjects without food intake may be more conductive to the establishment of GN-ARB compared with subjects with oral intake. Although PEG and NGs are effective for ensuring stable nutritional intake and preventing massive aspiration for patients with dysphagia, their role in preventing pneumonia is limited due to continued micro aspiration continues [28]. It is necessary to address the need for proper infection control not only for patients in acute hospitals but also for patients with parenteral intake due to the high possibility of ARB in their oral cavity.

In the intervention study, disappearance of oral ARB after the intervention was not observed in most subjects. Additionally, this study found that 8 of the 9 subjects who received GN-ARB before and after the intervention had the same GN-ARB bacterial species. Among the 11 GN-ARB-positive subjects before the intervention, GN-ARB was eliminated in only 3 subjects after the intervention, while GN-ARB was not isolated in 4 subjects after the intervention. In addition, a similar tendency regarding the isolation of *S. aureus* including MRSA after the intervention was observed. Since *S. aureus* is known to cause opportunistic infections, including pneumonia [29, 30], the elimination of *S. aureus* in the oral cavity is important for preventing aspiration pneumonia. Therefore, this study included *S. aureus* carriage as well as GN-ARB and MRSA in the interventional group. However, *S. aureus* elimination was also not observed after the intervention. Based on our findings, this study concluded that short-term professional oral management had a limited capacity to eliminate ARB. This study did not use disinfectants such as povidone iodine or chlorhexidine during the oral management. Therefore, the use of disinfectants in oral management may be considered effective for the elimination of ARB.

The facility used in our study has a hospital dental service, with part-time dentists and full-time hygienists providing oral management to residents, so the oral condition of the residents was constantly checked. Despite this environment, 51% of the subjects had ARB, and GN-ARB, MRSA and oral ARB were not eliminated by professional oral care. It was considered the presence of oral ARB to be related to systemic conditions, such as tube feeding, rather than to oral hygiene status. The subject who had pneumonia eight times had both oral and rectal ESBL-producing *E. coli,* used tube feeding, and died of refractory pneumonia. Older adults who required nursing care at level 4 or 5 were susceptible to infection due to decreased resistance. For example, if a subject harbored ARB and developed infections, the subject was difficult to treat. Therefore, this study suggested that avoidance of ARB infection is imperative, especially in subjects requiring a high level of care. An association has been found between tube feeding and the presence of GN-ARB, and care should be taken in the management of PEG tubes to assume the presence of ARB [5]. It has been reported that some GN-ARB isolated from the oral cavity do not respond to disinfectants; thus, the use of disposables is recommended [12].

The results of our cross-sectional and interventional studies suggest that elderly facilities, such as the facility in which our study was conducted, should be especially careful to avoid the transmission of ARB. To achieve this goal, it is important that all staff working in elderly care facilities have accurate knowledge regarding the presence of ARB in the oral cavity, the challenges associated with their elimination even with professional oral management and the necessity of implementing standard precautions.

## Conclusion

This study concluded that a short-term professional oral management has a limited capacity to eliminate ARB. It suggests that elderly facilities should be especially careful to avoid the transmission of ARB.

## Supporting information

**S1 Table. Primer information for PCR.** The primers used in this study are listed.
(XLSX)

**S2 Table. ARB colonization in the cross-sectional study.** The detail of ARB colonization from oral and rectal sample in the cross-sectional study are shown. For each subject, we summarized the bacterial species of ARB, the ESBL gene types and the number of different species of ARB which they had in the cross-sectional study.
(XLSX)

**S3 Table. Bacterial strains in the cross-sectional study.** We show the bacterial species, antimicrobial susceptibility and ESBL gene of bacterial strains (GN-ARB) isolated in the cross-sectional study.
(XLSX)

## Acknowledgments

We gratefully acknowledge the contributions and support of Tomomi Nakamura, Kanako Yamawaki, Tsugimi Yamane, Hiroko Tomiki, Sakiko Itaki, and all of the staff members of this facility. We also thank Spring Nature Author Services for English language editing.

## Author contributions

**Conceptualization:** Azusa Haruta, Mineka Yoshikawa, Maho Takeuchi, Miki Kawada-Matsuo, Mi Nguyen-Tra Le, Hitoshi Komatsuzawa.

**Data curation:** Azusa Haruta, Mineka Yoshikawa, Maho Takeuchi.

**Formal analysis:** Azusa Haruta, Mineka Yoshikawa.

**Funding acquisition:** Azusa Haruta, Mineka Yoshikawa, Hiroki Ohge, Motoyuki Sugai.

**Investigation:** Azusa Haruta, Mineka Yoshikawa, Maho Takeuchi, Miki Kawada-Matsuo, Toshiki Kajihara, Yo Sugawara, Junzo Hisatsune, Hitoshi Komatsuzawa, Motoyuki Sugai.

**Methodology:** Azusa Haruta, Mineka Yoshikawa, Mi Nguyen-Tra Le, Toshiki Kajihara, Hitoshi Komatsuzawa.

**Project administration:** Azusa Haruta, Mineka Yoshikawa.

**Resources:** Mineka Yoshikawa, Miki Kawada-Matsuo, Mi Nguyen-Tra Le, Hitoshi Komatsuzawa, Hiroki Ohge, Kazuhiro Tsuga.

**Supervision:** Hitoshi Komatsuzawa, Hiroki Ohge, Motoyuki Sugai, Kazuhiro Tsuga.

**Validation:** Azusa Haruta, Toshiki Kajihara, Hitoshi Komatsuzawa.

**Visualization:** Azusa Haruta, Mineka Yoshikawa, Hitoshi Komatsuzawa.

**Writing – original draft:** Azusa Haruta, Mineka Yoshikawa, Hitoshi Komatsuzawa.

**Writing – review & editing:** Maho Takeuchi, Miki Kawada-Matsuo, Hiroki Ohge, Motoyuki Sugai, Kazuhiro Tsuga.

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
