## [Decision Letter · Decision Letter 0]

26 Dec 2025

PONE-D-25-58602Oral and rectal colonization of Gram-negative resistant bacteria and Methicillin-resistant Staphylococcus aureusresidents in one long-term care facility and changes in professional oral hygiene care: Cross-sectional and interventional StudyPLOS One

Dear Dr. Haruta,

Thank you for submitting your manuscript to PLOS ONE. After careful consideration, we feel that it has merit but does not fully meet PLOS ONE’s publication criteria as it currently stands. Therefore, we invite you to submit a revised version of the manuscript that addresses the points raised during the review process.

We look forward to receiving your revised manuscript.

Kind regards,

Gabriel Trueba, PhD

Academic Editor

PLOS One

Journal Requirements:

This study was supported by the Ministry of Health, Labour, and Welfare Japan (program Grant No. JPMH19HA1004 and JPMH22HA1002)(OH), MEXT Grants-in-Aid for Scientific Research (C) (Grant No. 20K10248)(MY), and the Research Program on Emerging and Re-emerging Infectious Diseases from the Japan Agency for Medical Research and Development under Grant No. 21fk0108604j0001 and JP22fk0108604j0002(MS).

The authors have no conflicts of interest to declare.

Reviewers' comments:

Reviewer's Responses to Questions

**Comments to the Author**

1. Is the manuscript technically sound, and do the data support the conclusions?

Reviewer #1: Yes

Reviewer #2: Yes

2. Has the statistical analysis been performed appropriately and rigorously? 

Reviewer #1: Yes

Reviewer #2: Yes

3. Have the authors made all data underlying the findings in their manuscript fully available?

Reviewer #1: Yes

Reviewer #2: Yes

4. Is the manuscript presented in an intelligible fashion and written in standard English?

Reviewer #1: Yes

Reviewer #2: Yes

5. Review Comments to the Author

Reviewer #1: The authors described the oral and rectal colonization of antimicrobial resistant gram-negative bacteria and methicillin-resistant Staphylococcus aureus in residents of one long-term care facility and explore the effect of short-term professional oral management in controlling the colonizing resistant bacteria.

The article is well-written, and application of materials and methods are appropriate for this study.

This reviewer noticed the following points to be clarified.

1. Check the accuracy of the title.

2. Table 2 and Table 3, detection of CTX-M genes are group specific or specific CTX-M types?

Detection by PCR might be CTX-M- group specific, so that CTX-M1, CTX-M2, and CTX-M9 should be rephrased as CTX-M-group-1, CTX-M-group-2, and CTX-M-group-9 respectively.

Without sequencing of CTX-M gene covering the whole ORF, the specific CTX-M- type could not be assigned.

3. Table S1, references of primers are not listed in reference section. These references should be described in foot note of Table S1.

Reviewer #2: Greetings,

Very good work. Kindly consider the following points:

1- In this manuscript, the pronoun “we” has been used frequently (67 times). In scientific writing, it is preferable to avoid personal pronouns. Please replace them with formal expressions such as “this study,” “the present study,” or “the current study.”

2- Please add a clear Conclusion section following the Discussion section.

3- It is recommended to include a list of abbreviations at the end of the manuscript for better readability.

4- Several references are outdated. Kindly update the reference list by citing more recent studies published from 2018 onward.

Kind regards.

6. PLOS authors have the option to publish the peer review history of their article (what does this mean? ). If published, this will include your full peer review and any attached files.

**Do you want your identity to be public for this peer review?** For information about this choice, including consent withdrawal, please see our Privacy Policy .

Reviewer #1: No

Reviewer #2: No

---

## [Author Response · Author response to Decision Letter 1]

22 Jan 2026

Dear Editors and Reviewers

Thank you very much for reviewing our manuscript and offering valuable advice.

We have addressed your comments with point-by-point responses in 'Response to Reviewers', and revised the manuscript accordingly.

We would like you to confirm these files.

Best regards,

Haruta Azusa

---

## [Decision Letter · Decision Letter 1]

1 Feb 2026

Oral and rectal colonization of Gram-negative antimicrobial-resistant bacteria and Methicillin-resistant Staphylococcus aureus in one long-term care facility and changes in professional oral hygiene care: Cross-sectional and interventional study

PONE-D-25-58602R1

Dear Dr. Haruta,

We’re pleased to inform you that your manuscript has been judged scientifically suitable for publication and will be formally accepted for publication once it meets all outstanding technical requirements.

Kind regards,

Gabriel Trueba, PhD

Academic Editor

PLOS One

Additional Editor Comments (optional):

Reviewers' comments:

Reviewer's Responses to Questions

**Comments to the Author**

1. If the authors have adequately addressed your comments raised in a previous round of review and you feel that this manuscript is now acceptable for publication, you may indicate that here to bypass the “Comments to the Author” section, enter your conflict of interest statement in the “Confidential to Editor” section, and submit your "Accept" recommendation.

Reviewer #1: All comments have been addressed

Reviewer #2: All comments have been addressed

2. Is the manuscript technically sound, and do the data support the conclusions?

Reviewer #1: Yes

Reviewer #2: Yes

3. Has the statistical analysis been performed appropriately and rigorously? 

Reviewer #1: N/A

Reviewer #2: Yes

4. Have the authors made all data underlying the findings in their manuscript fully available?

Reviewer #1: Yes

Reviewer #2: Yes

5. Is the manuscript presented in an intelligible fashion and written in standard English?

Reviewer #1: Yes

Reviewer #2: Yes

6. Review Comments to the Author

Reviewer #1: The authors made changes according to the prior comments which has improved the clarity of the article.

No comments from this reviewer.

Reviewer #2: (No Response)

7. PLOS authors have the option to publish the peer review history of their article (what does this mean? ). If published, this will include your full peer review and any attached files.

**Do you want your identity to be public for this peer review?** For information about this choice, including consent withdrawal, please see our Privacy Policy .

Reviewer #1: No

Reviewer #2: No

---

## [Editor Report · Acceptance letter]

PONE-D-25-58602R1

PLOS One

Dear Dr. Haruta,

I'm pleased to inform you that your manuscript has been deemed suitable for publication in PLOS One. Congratulations! Your manuscript is now being handed over to our production team.

Kind regards,

on behalf of

Dr. Gabriel Trueba

Academic Editor

PLOS One